# Engineered fluoride sensitivity enables biocontainment and selection of genetically-modified yeasts

Justin I. Yoo[1], Susanna Seppälä [1] & Michelle A. O'Malley [1✉]

Biocontainment systems are needed to neutralize genetically modified organisms (GMOs) that pose ecological threats outside of controlled environments. In contrast, benign selection markers complement GMOs with reduced fitness. Benign selection agents serve as alternatives to antibiotics, which are costly and risk spread of antibiotic resistance. Here, we present a yeast biocontainment strategy leveraging engineered fluoride sensitivity and DNA vectors enabling use of fluoride as a selection agent. The biocontainment system addresses the scarcity of platforms available for yeast despite their prevalent use in industry and academia. In the absence of fluoride, the biocontainment strain exhibits phenotypes nearly identical to those of the wildtype strain. Low fluoride concentrations severely inhibit biocontainment strain growth, which is restored upon introduction of fluoride-based vectors. The biocontainment strategy is stringent, easily implemented, and applicable to several eukaryotes. Further, the DNA vectors enable genetic engineering at reduced costs and eliminate risks of propagating antibiotic resistance.

[1] Department of Chemical Engineering, University of California Santa Barbara, Santa Barbara, CA 93106, USA. ✉email: momalley@ucsb.edu

Rapid advancements in synthetic biology augment both our ability to engineer cellular functions as well as concerns associated with genetically modified organisms (GMOs). GMOs have been engineered to produce biofuels, chemicals, and pharmaceuticals at industrial scale[1–4], and the design and construction of microbial genomes[5–9] promise even greater capacity to engineer cells with precisely defined functions. However, these advances amplify concerns surrounding health and ecological risks posed by GMOs that house hazardous genetic material or have a fitness advantage over microbes found in natural ecosystems[10–12]. The potential release of GMOs is particularly concerning due to the emergence of do-it-yourself synthetic biology kits enabling construction of GMOs without the physical containment strategies present in academia and industry. In addition to biocontainment, most biotechnological applications would derive great benefit from benign selection markers as alternatives to antibiotics, which are costly and may incur risk of propagating antibiotic resistance through overuse of antibiotics and horizontal gene transfer (HGT) even from lysed cells[13–15].

While methods for biocontainment of bacteria have advanced rapidly[11,12], only two strategies have been demonstrated in the yeast Saccharomyces cerevisiae[16,17] despite their extensive use as production platforms in academia[18,19] and in industry[20,21]. Moreover, the two biocontainment strategies presented for S. cerevisiae require exogenous ligands and cellular machinery for survival and depend on inducible transcription of essential genes. This design strategy renders the safeguard mechanisms susceptible to inactivating mutations, which were indeed observed[16,17]. In contrast, a permissive state independent of mutable systems would markedly reduce the likelihood of biocontainment inactivation. Further, the ideal eukaryotic biocontainment strategy is compatible with various microorganisms. The generality of such a biocontainment system is increasingly important as non-model eukaryotic organisms continue to be developed as production platforms.

The mechanism underlying eukaryotic fluoride tolerance was recently elucidated in three eukaryotes and depends on the presence of at least one fluoride exporter protein, FEX1 or FEX2[22]. Recognizing the broad utility of this stringent selection mechanism, we sought to extend the application of fluoride sensitivity to two pressing needs in synthetic biology: biocontainment and alternative selection markers. Accordingly, we present a yeast biocontainment strain that is highly sensitive to fluoride and a complementary set of DNA vectors reliant on fluoride-based selection (Fig. 1).

## Results

**Biocontainment strain construction**. To sensitize yeast to fluoride, we previously generated a S. cerevisiae strain lacking both native fluoride exporter (FEX1/2) genes (Fig. 2a)[23]. Accordingly, the knockout strain (i.e., the biocontainment strain) is highly sensitive to fluoride exhibiting an IC$_{50}$ of 46.6 µM NaF, which is approximately three orders of magnitude lower than that of the wildtype (WT) parent strain (21.9 mM NaF) and agrees with the initial report[22] (Fig. 2b). Strikingly, growth of the biocontainment strain is severely inhibited by 210.5 µM NaF (Fig. 2a, b), which is equal to the U.S. EPA standard for drinking water quality[24] and significantly lower than concentrations observed in groundwater, where fluoride concentrations vary significantly depending on location and associated environmental factors[25]. While a review of >38,000 U.S. wells indicates that a majority (>80%) contain [F$^-$] <36.8 µM, many of these sites are proximal to those with high fluoride concentrations. Therefore, we speculate that intermixing of groundwater in regions with high [F$^-$] may increase the likelihood of growth inhibition of a

biocontainment strain released into nature. In addition to spatial variation, temporal fluctuation of fluoride concentration should be considered. As rainwater is poor in fluoride, surface water (e.g., lakes, rivers) and shallow groundwater often contain lower fluoride concentrations due to dilution by rain[26]. Similarly, areas with high rainfall can be expected to contain lower levels of fluoride. Thus, while fluoride sensitivity may serve as a robust biocontainment measure in areas known to have high fluoride concentrations, care should be taken to evaluate local fluoride abundance. Accordingly, fluoride-sensitivity enables passive biocontainment wherein cellular fitness is unperturbed under standard laboratory conditions and markedly reduced in nature where fluoride is in sufficient abundance. This strategy is inherently robust as its efficacy relies on the absence of endogenous genes rather than the presence and activity of essential genes, which are subject to continuous selection pressure and neutral drift.

**Biocontainment strain benchmarking**. To further benchmark our strain, we assessed the strain with focus on four factors that would describe an ideal biocontainment strategy[12]: (1) minimal fitness defects, (2) amenability to additional engineering, (3) escape rate below 1 in 10$^8$ cells, and (4) genetic robustness. In contrast to stringent selection in the presence of fluoride, minimal fitness defects are desired in the absence of fluoride. Indeed, colony morphology is identical to that of the WT parent strain in the absence of fluoride (Fig. 2a). Similarly, the growth rate of the biocontainment strain ($\mu_{max} = 0.54\,h^{-1}$) is nearly identical to that of WT ($\mu_{max} = 0.61\,h^{-1}$) (Fig. 2c). Further, phenotypic homogeneity and mean fluorescence intensity (MFI) of the two strains are nearly identical upon expression of yEGFP (Fig. 2d, e). The WT-level production of heterologous proteins is indicative of the biocontainment strain's capacity for additional engineering without pleiotropy. To determine the likelihood of biocontainment strain survival outside of controlled conditions, we determined the escape rate of our strain in the presence of 210.5 µM and 5 mM NaF. In both cases, the strain escape rate falls below the NIH guideline of 1 in 10$^8$ cells (Supplementary Table 1). In the presence of 5 mM NaF, the escape rate falls below the detection limit of our assay (1 in 10$^9$). While the observed escape rates reflect stringent growth inhibition in the presence of fluoride, these data do not necessarily reflect lethality. Relief of selection pressure could result in proliferation of the biocontainment strain. To assess whether fluoride is acting as a microbiostatic (i.e., growth inhibiting) or microbiocidal (i.e., lethal) agent, the biocontainment strain was grown in the presence of varying concentrations of fluoride, washed with sterile buffer, and used to inoculate fresh media. After 10 h, cultures previously exposed to fluoride only reached 13–39% of the control culture concentration (Fig. 2f). The WT strain was also treated with 16 mM fluoride prior to washing and resuspension in fresh media. Incubation of the WT strain with 16 mM fluoride appears to affect the strain's growth, albeit to a lesser extent compared to the biocontainment strain. Thus, fluoride appears to act as a microbiostatic agent under the examined conditions and reduces, but not entirely, cell viability after treatment. While the fluoride treatment reported here may not serve as a direct substitute for conventional sterilization techniques (e.g., autoclaving), reduced fitness will impair the strain's capacity to persist in nature supporting the use of fluoride sensitivity as a biocontainment measure.

Concerning the fourth characteristic, genetic robustness, growth of the biocontainment strain is carried out in the absence of selection pressure, greatly reducing the likelihood of generating revertants or evolved fluoride resistance. This stands in contrast

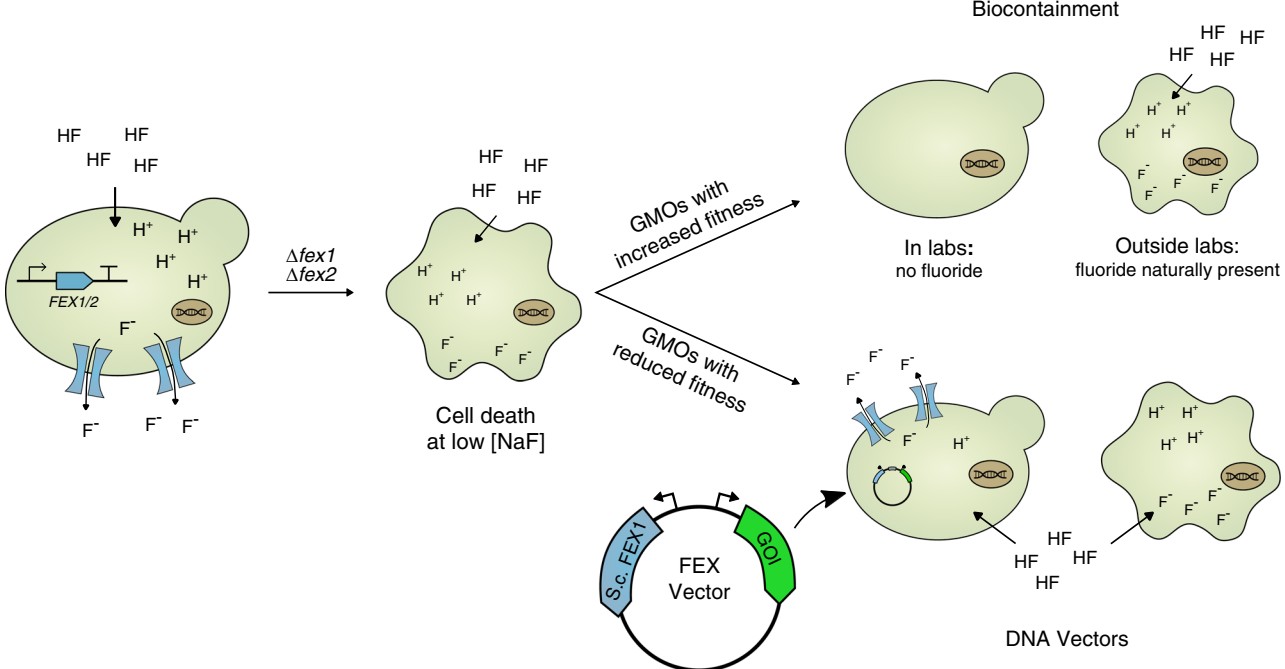

**Fig. 1 Engineered fluoride sensitivity augments biocontainment and selection systems.** Yeast cells lacking native fluoride exporter genes (*FEX1/FEX2*) are highly sensitized to low concentrations of fluoride yet retain wildtype phenotypes in the absence of the ion. Accordingly, this mechanism befits biocontainment of genetically modified organisms (GMOs) that pose ecological risks outside of laboratory environments. Alternatively, fluoride sensitivity can be leveraged to provide an alternative selection marker in GMOs with reduced fitness and likelihood of persistence in the environment.

to current yeast strategies[16,17] in which cells are subject to continuous selection pressure to maintain functional components of the biocontainment system or circumvent the system altogether. In theory, our biocontainment system can be neutralized upon HGT of a functional *FEX* gene cassette from an organism in nature. While horizontal transfer of bacterial genes to *S. cerevisiae* may have occurred in nature[27,28], a dedicated mechanism for free DNA uptake is yet to be discovered in *S. cerevisiae*[28]. Rather, bacterial, contact-dependent mechanisms (i.e., conjugation[29]) of DNA transfer are likely to have resulted in the presence of foreign genes in the *S. cerevisiae* genome. Accordingly, although possible, we posit that HGT of a *FEX* gene from a eukaryote is exceedingly unlikely to occur prior to death of the strain under selective pressure. Acquisition of fluoride tolerance could also be mediated through mating of the haploid biocontainment strain with a *FEX*-containing strain. However, a functional mating pathway is not necessary for cell viability, and yeasts are easily rendered incapable of mating through deletion of individual *ste* genes within the mating pathway[30].

**Fluoride sensitivity enables fluoride-based vector selection.** Complementary to a robust biocontainment system, cost-effective selection markers facilitate translation of lab-scale processes to the industrial scale. Maintenance of non-integrating plasmids or screening genomically integrated transformants using conventional selection agents are prominent operating costs at scale. In addition to elevating bioprocessing costs, the use of antibiotic selection introduces the risk of generating antibiotic-resistant microorganisms[13]. While auxotrophic selection eliminates the need for a selection agent, it necessitates the use of a defined medium, which can limit cellular growth and/or fitness. Further, in academic and research settings, there is a need for additional selection markers for genetic and metabolic engineering and synthetic biology. Exemplifying the need for alternative selection

markers, Novo Nordisk developed the *POT1* expression system, which enables production of insulin using *S. cerevisiae* in nutrient-rich media[31,32]. However, this system restricts the carbon source to glucose as the *POT1* marker restores the glycolytic pathway in *S. cerevisiae* strains harboring a mutated copy of the native *tpi*. This growth scheme precludes use of galactose-inducible promoters, which are often used to limit deleterious effects of heterologous gene products such as membrane proteins. To address these issues, we constructed a set of DNA vectors containing fluoride selection markers. By replacing auxotrophic selection markers with the *S. cerevisiae FEX1* gene in three commonly used vector backbones, we have constructed a set of yeast vectors that enable selective cell growth and production of heterologous proteins in rich, complex media containing low concentrations of NaF. At recommended working concentrations, conventional antibiotics used with *S. cerevisiae* cost between US $32 and $2175/L, while NaF costs US$0.04/L representing 3–5 orders of magnitude in potential savings (Supplementary Table 2) that become more prominent at industrial scales.

**Characterization of FEX vectors.** Introduction of a FEX vector into the knockout strain completely restores fluoride tolerance to WT levels (Fig. 3a). To challenge the robustness of the FEX vectors, we used the systems to express a human G protein-coupled receptor, the adenosine $A_2a$ receptor ($A_2aR$), as membrane protein expression often imparts metabolic burden by taxing the cell secretory pathway. Thus, we constructed integrating (pIFEX), centromeric (pCFEX1), and episomal (pEFEX1) vectors harboring $A_2aR$-*GFP* and *FEX* cassettes (Fig. 3b). In these vectors, constitutive expression of the *FEX* gene is driven by the *Ashbya gossypii* TEF1 promoter, which is commonly used to express auxotrophic markers in yeast plasmids. Upon expression of $A_2aR$-*GFP* from these plasmids, two distinct phenotypes emerge (Fig. 3c). Between 34% and 41% of cells harboring the non-integrating pCFEX1 or pEFEX1 constructs exhibit

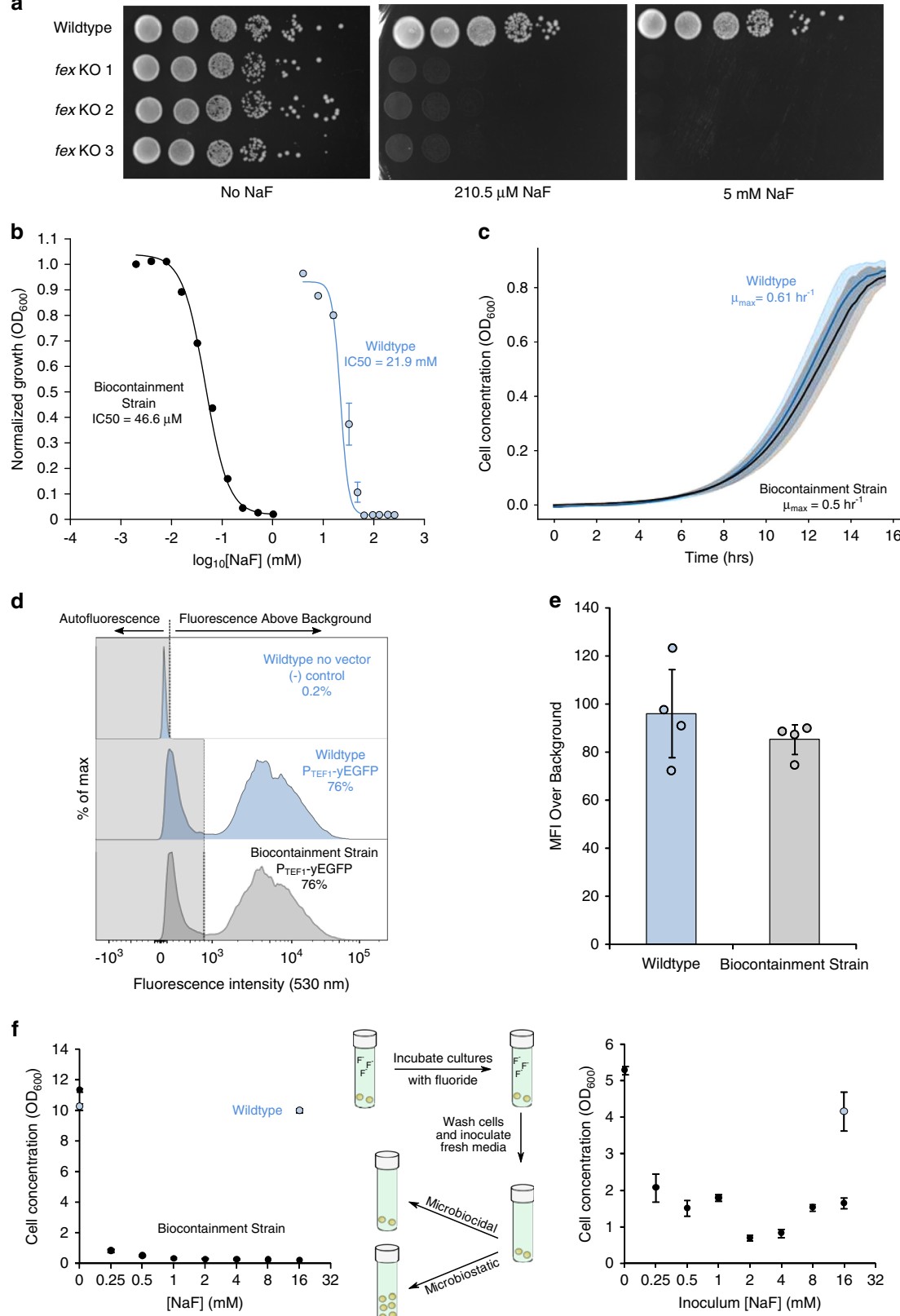

fluorescence intensities above autofluorescence. In contrast, 97% of cells harboring the genomically integrated pIFEX construct displays fluorescence above background. The FEX vectors display varying degrees of similarity to their auxotrophic counterparts characterized previously[33]. Expression of *A₂aR-GFP* from the high-copy backbones, pEFEX1 and pYES, yields similarly low

fluorescence intensities, and the majority of each population displays fluorescence intensities comparable to autofluorescence. In contrast, *A₂aR-GFP* expression from the auxotrophic low-copy vector, pYC, results in significantly greater MFI over background compared to the analogous FEX vector. Additionally, cells harboring pYC A₂aR-GFP exhibit a bimodal fluorescence

**Fig. 2 Yeast strains lacking *FEX* genes provide a stringent, passive biocontainment mechanism. a** A yeast spotting assay demonstrates sensitivity of yeast lacking *FEX* genes to μM concentrations of NaF. **b** Dose response curve of the biocontainment strain illustrates three orders of magnitude greater sensitivity to fluoride compared to the parent wildtype strain. **c** In the absence of fluoride, growth of the biocontainment and wildtype parent strains in YPD is nearly indistinguishable. **d**, **e** Phenotypes of yEGFP production in the biocontainment and wildtype strains are also nearly identical, demonstrating the versatility of the biocontainment strain for engineering without deleterious effects associated with *FEX* knockouts. **f** Cell growth is severely inhibited both in the presence of NaF as well as after washing and subculturing into fresh media, albeit to a lesser extent. In **b**, **c**, **e**, and **f**, data represent the mean of three biological replicates, and error bars represent their standard deviation. The histograms presented in **d** correspond to representative samples. Source data are provided as a Source Data file.

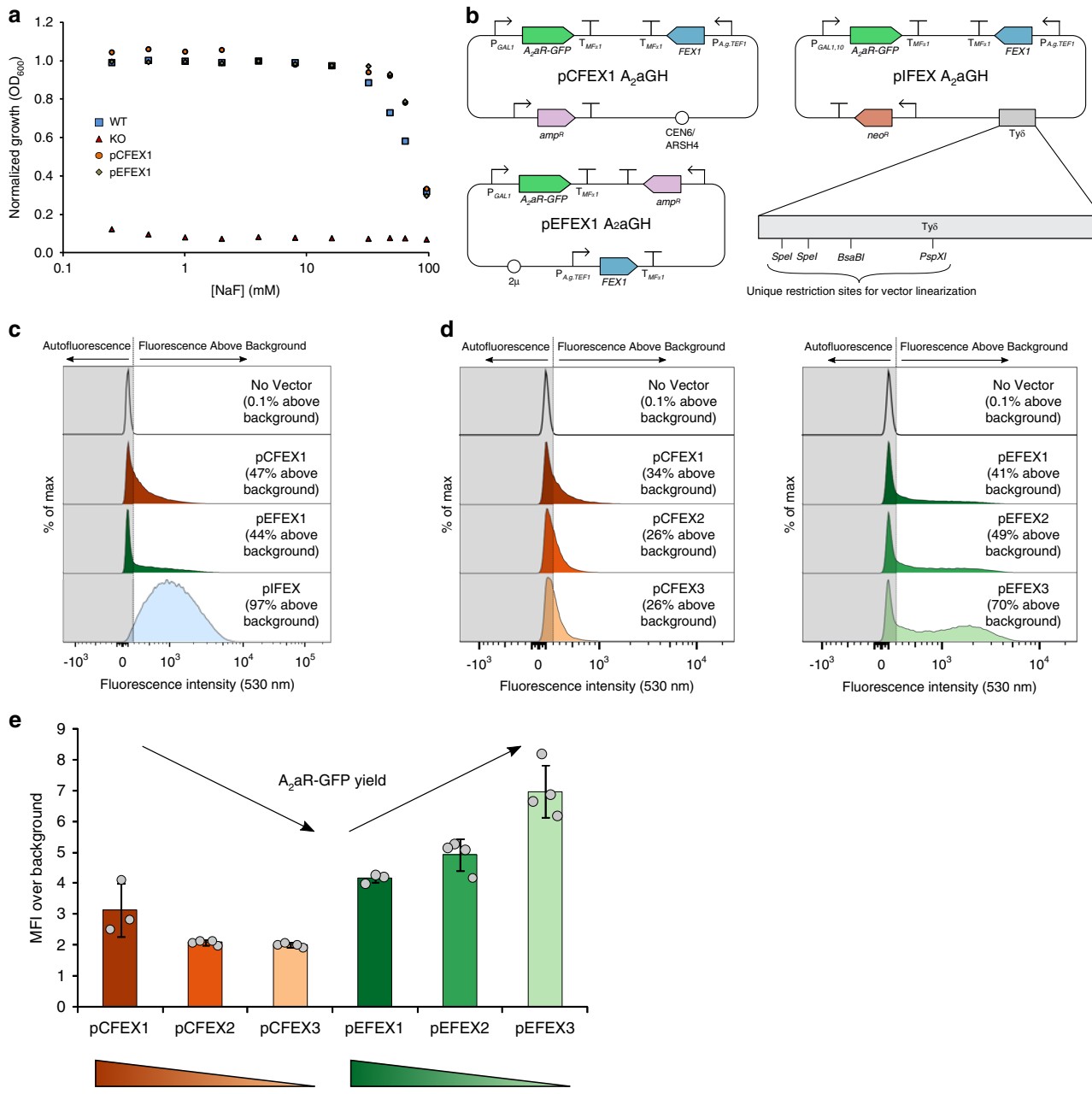

**Fig. 3 Fluoride-sensitive yeast strains enable use of *FEX* as a selection marker. a** While growth of the FEX knockout (KO) strain is inhibited by μM concentrations of fluoride, transformation with a FEX-based vector restores fluoride resistance to wildtype levels. **b** Diagrams of centromeric (pCFEX1), episomal (pEFEX1), and integrating (pIFEX) vectors. **c** FACS analysis of *A₂aR-GFP* expression from pCFEX1 and pEFEX1 reveals that a significant portion of cells exhibit background fluorescence intensities. In contrast, expression from the integrating pIFEX backbone yields a unimodal fluorescence distribution where 97% of the population displays fluorescence above background. **d**, **e** Replacement of the promoter driving *FEX1* expression significantly influences A₂aR-GFP yield and phenotypic homogeneity in pEFEX but has less pronounced effects on pCFEX. In **c** and **d**, histograms correspond to representative samples. Data present in **a** represents a single replicate for each strain. Data presented in **e** represents the mean of three biological replicates, and error bars represent their standard deviation. Source data are provided as a Source Data file.

distribution, whereas a unimodal distribution is associated with pCFEX1 A$_2$aR-GFP where the majority of cells exhibit fluorescence intensities indistinguishable from autofluorescence. In all, A$_{2a}$R-GFP expression from the FEX vectors was initially poorer than expression from their auxotrophic counterparts.

**Optimization of expression from FEX vectors**. We sought to improve A$_2$aR-GFP yields associated with the non-integrating vectors, and initially hypothesized that plasmid loss contributes to the large fraction of cells displaying basal fluorescence intensity as observed previously[33]. Non-integrating plasmid loss is often a result of low mitotic stability. In other words, as a cell undergoes mitotic cell division, a non-integrating plasmid is more likely to be lost from the cell than one integrated into the genome. As demonstrated using the partially impaired *LEU2-d* and *URA2-d* selection markers, plasmid copy number can be modulated to cope with an imposed selection pressure[34,35]. Instead of modifying the *FEX* marker, we sought to impose stronger selection pressure by increasing the exogenous fluoride concentration. Fluoride transport across cellular membranes occurs primarily through diffusion of hydrogen fluoride (HF)[36]; therefore, we generated a simple model of HF transport across the cell membrane, which indicates that a decrease of 1 pH unit at 10 mM NaF produces a tenfold increase in fluoride flux (Supplementary Fig. 1). However, marginal differences in protein yield and phenotypic homogeneity are observed upon gene expression at lower pH (Supplementary Fig. 2).

Next, we hypothesized that the metabolic burden associated with producing two membrane proteins results in plasmid loss due to stress placed on the secretory pathway. This hypothesis is supported by the phenotype associated with the mitotically stable, integrating pIFEX vector. Therefore, we replaced the promoters driving *FEX* expression in pCFEX1 and pEFEX1 with weaker constitutive promoters (Supplementary Fig. 3). Upon replacing the *A. gossypii* TEF1 promoter in pEFEX1 with the weaker PGI1 (ref. [37]) (pEFEX2) and REV1 (ref. [38]) (pEFEX3) promoters, the proportion of cells exhibiting fluorescence intensity above background increased to 49% and 70%, respectively (Fig. 3d). Total A$_2$aR-GFP yield also increases appreciably upon gene expression in pEFEX2 and pEFEX3 (Fig. 3e). Notably, expression of *A$_{2a}$R-GFP* from pEFEX3 yields a distinct subpopulation of cells exhibiting high fluorescence intensity resulting in a bimodal distribution (Fig. 3d). This phenotype represents a marked improvement compared to that observed using the auxotrophic high-copy vector described previously[33]. In contrast, promoter swapping in the pCFEX1 backbone did not yield appreciable differences in expression patterns (Fig. 3d) and even led to a reduction in A$_2$aR-GFP yield (Fig. 3e). We speculate that the differences between the pCFEX and pEFEX systems arise due to differences in vector copy numbers. As pEFEX is likely maintained in higher copies, reduced *FEX1* expression effectively reduces metabolic burden, which is common to secreted and membrane protein production from high-copy vectors[33,39,40]. In contrast, A$_2$aR-GFP production from the low-copy pCFEX vectors may not sufficiently tax the yeast secretory pathway to gain benefit from reduced *FEX1* expression. Instead, lower FEX1 yields may increase sensitivity to exogenous fluoride leading to increased cell death.

## Discussion

This work leverages fluoride sensitivity in biocontainment and plasmid systems. Our biocontainment strain offers several advantages over existing strategies. First, our strain is immediately amenable to academic and industrial applications as growth media is prepared using ultrapure water lacking fluoride.

Although residual fluoride is present in certain reagents, ~7 μM in YPD media[22], the biocontainment strain showed no fitness defects compared to a WT control. Second, the knockouts can be easily introduced into any *S. cerevisiae* strain using accessible genetic engineering techniques such as CRISPR or *Delitto Perfetto*[41]. These gene deletions can also be made directly in a previously modified organism to multiplex safeguards or introduce a new safeguard into an existing GMO. Third, in areas where fluoride is naturally abundant in the environment[42], escapees are subject to lethal conditions without manual intervention. Despite the ubiquity of fluoride in the environment, laboratory conditions are stringently maintained. Thus, environmental fluoride is unlikely to contaminate bioreactors. Finally, other eukaryotic organisms containing fluoride transporter gene deletions exhibit sensitivity to fluoride[22]; thus, it is likely that our biocontainment strategy can be extended to a wide range of organisms relevant to biotechnology. The potential contamination of cultures by WT yeast represents a disadvantage of the FEX vector system, which is shared by auxotrophic selection strategies. While best laboratory practices will reduce the likelihood of contamination, FEX vectors can be used in combination with auxotrophy to minimize contaminations risks at the cost of precluding use of nutrient-rich, complex media.

In areas where fluoride is naturally abundant, the biocontainment platform is useful to reduce the fitness of GMOs that would otherwise prosper upon release into the environment; however, most genetic modifications will reduce an organism's ability to outcompete native microbes. Thus, we constructed a set of vectors that complements the Δ*fex* background conferring fluoride tolerance to maintain heterologous DNA. While engineered fluoride sensitivity was previously used as the basis of a selection marker in a CRISPR/Cas system in *Schizosaccharomyces pombe*[43], we extend the use of fluoride sensitivity to enable heterologous expression of any gene in the various DNA backbones available to the model yeast, *S. cerevisiae*. In contrast to auxotrophic markers, the FEX vectors enable use of rich media, which will not limit the growth or metabolism of engineered strains. Fluoride-based selection drastically reduces the cost of selection compared to antibiotics and precludes risks of generating antibiotic resistance. We expect that our contributions will be of immediate use in both academic and industrial settings to advance efforts in synthetic biology.

## Methods

**Plasmid construction**. In non-integrating yeast vectors, auxotrophic markers are often placed under the control of the *A. gossypii* P$_{TEF1}$ promoter. Thus, we first constructed an *A. gossypii* P$_{TEF1}$-*S. cerevisiae FEX1*-T$_{MFα1}$ cassette in the pITy backbone through two steps. First, primers 1 and 2 were used to append *EcoRI* and *EagI* sites to an *A. gossypii* P$_{TEF1}$ fragment amplified from pSVA13 (Supplementary Table 3). This fragment and pITy A$_2$aGH were digested with *EcoRI-HF* and *EagI-HF* and ligated to generate an intermediate pITy *A. gossypii* P$_{TEF1}$ A$_2$aGH construct. Second, primers 3 and 4 were used to amplify *S. cerevisiae FEX1* from BJ5465 gDNA extracted using the protocol provided by Lõoke et al.[44]. This fragment and pITy *A. gossypii* P$_{TEF1}$ A$_2$aGH were digested with *EagI-HF* and *AflII* and ligated to generate pITy *A. gossypii* P$_{TEF1}$-*S. cerevisiae FEX1*-T$_{MFα1}$. USER[45] cloning was used to subclone the *A. gossypii* P$_{TEF1}$-*S. cerevisiae FEX1*-T$_{MFα1}$ cassette, which was amplified using primers 5 and 6, into the pYC2/CT and pYES2 backbones amplified using primer pairs 7 and 8 and 8 and 9, respectively. The resulting constructs were named pCFEX1 A$_2$aGH and pEFEX1 A$_2$aGH, respectively. The *A. gossypii* P$_{TEF1}$ promoters were swapped with P$_{PGI1}$ and P$_{REV1}$ promoters amplified from BJ5465 gDNA using primer pairs 10 and 11 and 12 and 13, respectively. pCFEX1 A$_2$aGH and pEFEX1 A$_2$aGH backbones were amplified using primer pairs 7 and 14 and 9 and 14, respectively, to mediate promoter swapping through USER cloning. The pCFEX2 and pEFEX2 backbones carry the P$_{PGI1}$ promoter, and the pCFEX3 and pEFEX3 backbones carry the P$_{REV1}$ promoter. The integrating pIFEX A$_2$aGH construct was generated through USER cloning after amplifying pITy A$_2$aGH using primers 15 and 16 and the *A. gossypii* P$_{TEF1}$-*S. cerevisiae FEX1*-T$_{MFα1}$ cassette using primers 5 and 6. The pIFEX A$_2$aGH construct was designed to retain the *NEO* CDS to facilitate cloning in *Escherichia coli* using kanamycin; thus, the vector also confers G418 resistance to yeast. All constructs

were sequence-verified using Sanger sequencing (Genewiz) and transformed into *S. cerevisiae* using the high-efficiency lithium acetate protocol[46]. We found that a recovery period is necessary to obtain yeast transformants using the FEX vectors. Following resuspension in YPD, transformed yeast cells were incubated at 30 °C for at least 3 h prior to plating on YPD plates containing 210.5 μM NaF. Yeast transformed with pIFEX A₂aGH were plated on YPD supplemented with 10 mM NaF to promote increased gene dosage. The pRS315 P$_{TEF1}$-yEGFP construct was generated from pRS315 P$_{TEF1}$-yEGFP-Cln2, which was cloned for a separate study, in a series of steps. First, the pRS315 backbone was digested with *EagI*, blunted with Klenow fragment, and digested with *SpeI*. The P$_{TEF1}$-yEGFP-Cln2 cassette was amplified from YEp351 P$_{TEF1}$-yEGFP-Cln2 using universal M13 forward and reverse primers, then the amplicon was digested with *SpeI* to mediate directional cloning into pRS315. The backbone and insert were ligated and transformed into *E. coli*. Next, USER cloning mediated construction of pRS315 P$_{TEF1}$-yEGFP from the Cln2-tagged plasmids. Primers 17 and 18 were used to amplify the plasmid excluding the Cln2 tag and introducing two stop codons at the 3′-end of the yEGFP coding sequence.

**Yeast strains and culturing conditions.** *S. cerevisiae* strain BJ5465 (Mat**a** *ura3-52 trp1 leu2Δ1 his3Δ200 pep4::HIS3 prbΔ1.6R can1*) (ATCC) was used to construct the biocontainment strain BJ5465 *fex1::GSHU Δfex2* using the *Delitto Perfetto* method[47]. Primers 19 and 20 were used to integrate the GSHU cassette at the *FEX1* locus, and primers 19 and 21 were used to integrate the CORE-Kp53 cassette into the *FEX2* locus. Subsequently, primers 22 and 23 were used to remove the CORE-Kp53 cassette. Culture maintenance and gene expression were carried out using YPD medium at 30 °C with shaking at 225 r.p.m. Cultures harboring pCFEX and pEFEX plasmids were maintained in YPD supplemented with 2 mM NaF.

**Fluorescence-activated cell sorting.** Yeast cultures were diluted to an OD$_{600}$ = 1.0 in 1× phosphate buffered saline (PBS) prior to all fluorescence-activated cell sorting (FACS) analyses. Approximately 60,000 cells were analyzed from each sample using a 488-nm laser and 530/30 nm bandpass filter using the gating strategy illustrated in Supplementary Fig. 4. All analyses were conducted using a BD FACSAria I flow cytometer and FlowJo v10. To analyze yEGFP expression, WT and biocontainment strains harboring pRS315 P$_{TEF1}$-yEGFP were used to inoculate 5 mL synthetic dextrose medium supplemented with amino acids lacking leucine (SD -leu)[47]. Following overnight growth, cells were resuspended in 1× PBS and analyzed using FACS as described above. To analyze A₂aR-GFP expression from pCFEX and pEFEX vector backbones, knockout strains carrying the vectors were first cultured overnight in YPD medium supplemented with 2 mM NaF at 30 °C with shaking at 225 r.p.m. Following overnight growth, each culture was sub-cultured into YP medium supplemented with 2% (w/v) raffinose (YPR) and 2 mM NaF at an initial OD$_{600}$ of 0.5. Following ~10 h of shaking at 30 °C, each culture was subcultured into YP medium supplemented with 2% (w/v) raffinose, 2% (w/v) galactose (YPRG), and 2 mM NaF to induce A₂aR-GFP expression. Cultures were incubated with shaking at 30 °C overnight prior to flow cytometric analysis. Analysis of A₂aR-GFP expression from the pIFEX backbone was accomplished using a similar induction scheme in the absence of NaF. Knockout strains harboring integrated pIFEX A₂aR-GFP cassettes were cultured in YPD medium overnight at 30 °C with shaking at 225 r.p.m. Subsequently, cultures were subcultured into YPR medium and incubated at 30 °C with shaking. After ~10 h, A₂aR-GFP expression was induced in each culture through subculturing into YPRG medium and incubation at 30 °C with shaking overnight.

**Dilution spotting.** Yeast cultures were grown to an OD$_{600}$ ~ 3 prior to dilution to an OD$_{600}$ = 2.5 in sterile YPD. Diluted cells were used to prepare serial dilutions up to $10^{-5}$ in tenfold increments. A total of 5 μL of each dilution was spotted onto solid media using a multichannel pipette. Plates were allowed to dry at room temperature prior to overnight incubation at 30 °C.

**Fluoride dose response assay.** WT and biocontainment strain cultures were grown in biological triplicate overnight in YPD at 30 °C with shaking at 225 r.p.m. In the morning, the cultures were used to inoculate 5 mL fresh YPD at an initial OD$_{600}$ of 0.15. Cultures were incubated with shaking at 30 °C for 7 h, reaching OD$_{600}$ values near 2, and used to inoculate 3 mL YPD in individual wells of a 24-well block (Qiagen #19583) containing serially diluted concentrations of NaF and covered with a Breathe Easier sealing membrane (Sigma-Aldrich Z763624). Following overnight shaking at 30 °C, OD$_{600}$ values were measured for cultures in each well.

**Growth curves.** To generate growth curves, the WT and biocontainment strains were used to inoculate 5 mL YPD cultures, which were grown overnight at 30 °C with shaking at 225 r.p.m. Cultures were used to inoculate 1 mL YPD in individual wells of a 24-well plate (Corning 3526), at an initial OD$_{600}$ = 0.02. Cell growth was monitored using a Tecan Spark microplate reader maintained at 30 °C with orbital shaking at 180 r.p.m. and 3 mm amplitude. OD$_{600}$ measurements were taken every 10 min with 50 ms settling time prior to each reading. Specific growth rates were

calculated by fitting data to the logistic function[48] (Eq. (1)):

$$N(t) = N_0 + \frac{N_{asymp} - N_0}{1 + e^{[k(t_c - t)]}}, \tag{1}$$

where $N_0$ is the initial number of cells, $N_{asymp}$ is the maximal number of cells approached during stationary phase, $k$ is the growth rate, and $t_c$ is the time at which the growth curve exhibits an inflection point.

**Escape rate determination.** To determine the escape rate, the biocontainment strain was grown overnight in biological triplicate in YPD at 30 °C with shaking at 225 r.p.m. In the morning, ~50 colony forming units (CFUs) of each replicate culture were plated onto YPD media assuming a conversion factor of $10^7$ CFU/mL/ OD$_{600}$. Using the same conversion factor, $10^8$ and $10^9$ CFUs of each replicate culture were plated onto YPD supplemented with 210.5 μM and 5 mM NaF. Following incubation of plates at 30 °C for 2 days, CFUs were counted. The CFU values obtained for the YPD control plates were used to correct the CFU/mL/ OD$_{600}$ conversion factor and to calculate the total number of cells plated on each plate.

**pH Buffering experiment.** Individual colonies were used to inoculate 5 mL YPD medium. Cultures were grown overnight at 30 °C with shaking at 225 r.p.m. Following overnight growth, cells were used to inoculate YPR at an initial OD$_{600}$ = 0.2. Cultures were grown for 7–8 h at 30 °C with shaking at 225 r.p.m. and used to inoculate 5 mL YPRG supplemented with 10 mM NaF at an initial OD$_{600}$ = 0.2. One set of cultures were used to inoculate 5 mL YPRG buffered to pH = 6 using 100 mM MES (Sigma-Aldrich, St. Louis, MO, USA). Following overnight growth, cells were prepared for FACS analysis as described above.

**Microbiostatic/microbiocidal assay.** WT and biocontainment strain cultures were grown overnight and subcultured into 5 mL YPD at an initial OD$_{600}$ of 0.1. Cultures were incubated with shaking at 30 °C for ~10 h to an OD$_{600}$ between 2 and 4 and used to inoculate 3 mL YPD in individual wells of a 24-well block containing serially diluted concentrations of NaF and covered with a Breathe Easier sealing membrane. Following overnight incubation with shaking at 30 °C, the OD$_{600}$ values were measured for cultures in each well. Following growth, 0.2 OD-mL of cells were spun at $3000 \times g$ for 30 s, washed with sterile 1× PBS, and used to inoculate 3 mL fresh YPD in a sterile well of a 24-well block, which was covered with a Breathe Easier sealing membrane. Upon overnight growth, the OD$_{600}$ values were measured for cultures in each well.

**Modeling cellular fluoride uptake.** Fluoride uptake was approximated by modeling the transport of HF across the cell membrane. First, the concentration of HF in bulk is calculated from the exogenous NaF concentration using Eq. (2):

$$[\mathrm{HF}] = \frac{[\mathrm{NaF}]}{\left(\frac{10^{-3.17}}{10^{-pH}}\right) + 1}. \tag{2}$$

The above equation takes into account the pKa of HF at 25 °C, which is equal to 3.17. Next, the general transport equation (Eq. (3)) can be solved for the time-dependent concentration of NaF inside of the cell:

$$\frac{\partial C}{\partial t} = D\nabla^2 C, \tag{3}$$

where $C$ is the time-dependent concentration of fluoride and $D$ is the diffusion coefficient of fluoride. Solving Eq. (3) for $C$:

$$C_i(t) = C_0\left(1 - e^{-\frac{AP}{V}t}\right), \tag{4}$$

where $C_i(t)$ is the intracellular fluoride concentration, $C_0$ is the bulk fluoride concentration, $A$ is the membrane surface area, $P$ is the permeability constant, and $V$ is the cell volume. The permeability constant used for HF in the cell membrane is 0.0002 cm/s as calculated by Gutknecht et al.[49]. The cell surface area and volume were estimated from figures provided on bionumbers.hms.harvard.edu.

Now, the fluoride concentrations can be used to solve for the flux, $J$, of fluoride across the cell:

$$J = -P(C_i - C_0), \tag{5}$$

$$J = PC_0 e^{-\frac{AP}{V}t}. \tag{6}$$

Now, Eq. (6) gives the flux of fluoride across the cell membrane given a bulk concentration of fluoride, which is dictated by the exogenous NaF concentration as calculated in Eq. (2).

**Reporting Summary.** Further information on research design is available in the Nature Research Reporting Summary linked to this article.

## Data availability
The data that support the findings of this study are available in this manuscript, the supplementary materials, and from the corresponding author upon request. The

biocontainment strain, BJ5465 fex1::GSHU Δfex2, will be made available via reasonable request to M.A.O. The pIFEX, pEFEX, and pCFEX vectors and associated vector maps described in this work are available from Addgene (deposit number 78647). Source data are provided with this paper.

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

## Acknowledgements

We thank Dr. Stephen Streatfield for helpful discussions regarding biocontainment systems and Professor Simon Avery for pSVA13. J.I.Y. acknowledges support from a National Science Foundation Graduate Research Fellowship under grant no. 1650114. The authors further acknowledge funding support from the National Science Foundation (MCB-1553721).

## Author contributions

J.I.Y. and M.A.O. conceived the study, designed the experiments, analyzed the data, and wrote the manuscript. J.I.Y. and S.S. conducted the experiments.

## Competing interests

The authors declare the following competing interests: J.I.Y. and M.A.O. are authors on a patent application, application number 63072933, which has been filed. All other authors declare no competing interests.
