## [Peer Review File · Nature Communications]

Reviewers' Comments:

Reviewer #1:

Remarks to the Author:

The authors have addressed some of my previous comments and the manuscript has improved from the last version. It is well done, and exciting. I do however still have some concerns for both the biocontainment section as well as for the expression plasmid section that should be addressed. My main point is that I am still not convinced about the usefulness of strategy as a biocontainment system, and so would ask the authors to soften these arguments and address the concerns raised below. In contrast, however, I can see the point of using it as a selection marker for *S. cerevisiae*. However, it would benefit from a more direct comparison to other *S. cerevisiae* expression systems and I hope the authors add those control experiments to the manuscript. Below, I highlight my major concerns.

Concerns:

Lines 68-80: I don't agree with the statement that '210.5 μM F^- is significantly lower than concentrations observed in groundwater'. It looks to me that most ground water contains $[\text{F}^-] < 36.8 \mu\text{M}$, which might even be more diluted during rainfall. Amini et al. (10.1021/es071958y) report a median concentration across the USA of 7 μM F^- in ground water with only 0.3% of samples exceeding the WHO limit of 79 μM NaF . These numbers make it hard to believe that the current strain can be successfully contained using the presented strategy. Can the authors show any evidence that many of the >38,000 U.S. wells are proximal to ones with high fluoride concentrations? The data presented in the paper the authors cite (<https://doi.org/10.1016/j.scitotenv.2020.139217>) seems to show the opposite to me, with a vast area falling below 36.8 μM F^- . With an abundance of readily available data arguing against the ability of this system to contain an organism based on the F^- levels actually present in groundwater, it seems inappropriate to continue to assert that it could. If I am misunderstanding the data, please help me understand.

Line 90: I don't think that mentioning the maximum NaF concentration measured is very helpful here and would prefer the authors talk about the media.

Line 93: Figure 2f should be extended to show OD(600) for 100, 50 and 25 μM to show the relevant range of ground water fluoride concentrations.

Figure 2f: In their response to my previous question about the difference in WT OD between the left and the right panel the authors state:

"The WT strain was also treated with fluoride (16 mM) prior to washing and resuspension in fresh YPD. Overnight incubation of the WT strain with this concentration of fluoride appears to affect the strain's growth, albeit to a lesser extent compared to the knockout strain."

That is an important observation and should be stated in the manuscript as well.

Line 109: Strain escape rate should also be evaluated for 100, 50 and 25 μM to show the relevant

range of ground water fluoride concentrations.

Line 183: FACS signal for the auxotrophic vectors should be included as a positive control and direct comparison (especially pYC). Is the improved signal for pEFEX3 comparable to pYC? Or is pYC still the better option? Figure 3e should also contain a bar for pYC for direct comparison.

Line 205: To my previous question the authors state:

"While we agree that the low copy pCFEX plasmid system was not successfully optimized for A2aR-GFP production in this study, we stress that membrane proteins are particularly challenging to produce in a heterologous host. We chose this protein to challenge the FEX plasmid systems, and we find that expression of A2aR-GFP from each vector results in increased MFI compared to a negative control (Fig.3c), which suggests that the selection marker is functional, albeit to an suboptimal extent."

It would be helpful to see the results for a less stringent/taxing expression, maybe just GFP alone. This would help convincing me that the low copy number plasmid works and that the unsuccessful experiment with A2aR-GFP is due to the complex nature of the POI.

Line 238: Abundant in the environment at inhibiting concentrations?

Reviewer #2:
Remarks to the Author:

Summary:

This manuscript is a revised version of one that I recently reviewed for a sister journal. The authors have satisfactorily addressed the major and minor critiques that I described in my original review. I recommend the paper for publication in Nature Communication.

Reviewers' comments (marked in red) with Author's response (marked in black):

Reviewer #1 (Remarks to the Author):

The authors have addressed some of my previous comments and the manuscript has improved from the last version. It is well done, and exciting. I do however still have some concerns for both the biocontainment section as well as for the expression plasmid section that should be addressed. My main point is that I am still not convinced about the usefulness of strategy as a biocontainment system, and so would ask the authors to soften these arguments and address the concerns raised below. In contrast, however, I can see the point of using it as a selection marker for *S. cerevisiae*. However, it would benefit from a more direct comparison to other *S. cerevisiae* expression systems and I hope the authors add those control experiments to the manuscript. Below, I highlight my major concerns.

We thank the reviewer for their comments, especially now through two rounds of reviews of this manuscript. The authors have carefully considered the comments of the reviewer – particularly to address whether the implementation of new experiments would alter the main message or impact of the publication. We are delighted that the reviewer has identified our work as “well done, and exciting”, particularly how the fluoride selection can be used as a widespread selection system for *S. cerevisiae* and beyond. We recognize that the reviewer has some lingering concerns for the utility of this biocontainment system, particularly benchmarked against other existing approaches (as detailed below). We appreciate and recognize these limitations, and have therefore softened the language in the manuscript when discussing the impact of the biocontainment system in low fluoride concentration environments, and for the expression of different proteins of interest (POIs) (detailed responses below). Additionally, we have referenced previous studies to answer questions related to FACS signals for different auxotrophic vectors. The authors believe that the edited points and language in the revised manuscript are sufficient to address these concerns, and that the outcome of new proposed experiments (if conducted) would not alter the major take-home messages of the manuscript.

Additionally, we regret to report that our lab is only able to accommodate 1 person per day at UCSB under current COVID-19 safety restrictions. Given that our laboratory hosts 20+ researchers that are all competing for access to our lab spaces, implementing the suggested experiments would likely take several months at the sacrifice of several other priorities. Given that the PI is responsible for furthering many concurrent PhD projects and balancing the needs of all trainees during this difficult time, it is difficult to justify prioritizing lab access to complete the proposed experiments here – especially as we believe they would not change the direction or message of the paper as now written. We therefore respectfully ask the reviewer to consider these limitations when considering this resubmitted and revised manuscript.

Concerns:

Lines 68-80: I don't agree with the statement that '210.5 μM F⁻ is significantly lower than concentrations observed in groundwater'. It looks to me that most ground water contains [F⁻] < 36.8 μM , which might even be more diluted during rainfall. Amini et al. (10.1021/es071958y) report a median concentration across the USA of 7 μM F⁻ in ground water with only 0.3% of samples exceeding the WHO limit of 79 μM NaF. These numbers make it hard to believe that the current strain can be successfully contained using the presented strategy. Can the authors show any evidence that many of the >38,000 U.S. wells are proximal to ones with high fluoride concentrations? The data presented in the paper the authors cite (<https://doi.org/10.1016/j.scitotenv.2020.139217>) seems to show the opposite to me, with a vast area falling below 36.8 μM F⁻. With an abundance of readily available data arguing against the ability of this system to contain an organism based on the F⁻ levels actually present in groundwater, it seems inappropriate to continue to assert that it could. If I am misunderstanding the data, please help me understand.

We thank the Reviewer for their thoughtful review of our manuscript. While we appreciate their comments, we respectfully disagree with the Reviewer's assessment of the utility offered by the biocontainment strain. We believe our platform adds an important tool to the growing synthetic biology toolkit. As we demonstrate in Fig. 2b, the biocontainment strain exhibits an IC50 of 46.6 μ M fluoride. This result suggests that this strain's growth may be inhibited, albeit not completely, by a large number of natural water sources. Recognizing the stringent requirements of a biocontainment platform, we do state on lines 76 - 80,

*“Thus, while fluoride sensitivity may serve as a robust biocontainment measure in areas known to have high fluoride concentrations, **care should be taken to evaluate local fluoride abundance**. Accordingly, fluoride-sensitivity enables passive biocontainment wherein cellular fitness is unperturbed under standard laboratory conditions and markedly reduced **in nature where fluoride is in sufficient abundance**.”*

Additionally, we posit on lines 238 – 240 that the multiplexing of biocontainment strategies is an attractive approach to reduce the drawbacks of individual biocontainment mechanisms,

“These gene deletions can also be made directly in a previously modified organism to multiplex safeguards or introduce a new safeguard into an existing GMO.”

As construction of the biocontainment strain is quite straightforward (two gene deletions), we believe that the ease of introducing our biocontainment mechanism to a cell strain is an advantage of our platform.

We also note that the sole reports, two in total, presenting methods for *S. cerevisiae* biocontainment also demonstrate weaknesses as noted on lines 38 – 42,

*“...the two biocontainment strategies presented for *S. cerevisiae* require exogenous ligands and cellular machinery for survival and depend on inducible transcription of essential genes. This design strategy renders the safeguard mechanisms susceptible to inactivating mutations, which were indeed observed^{16,17}. In contrast, a permissive state independent of mutable systems would markedly reduce the likelihood of biocontainment inactivation.”*

The combination of the existing biocontainment platform with the one presented in this manuscript may result in a synergy that proves to be robust to varying environmental conditions. Accordingly, while we do believe the fluoride-sensitive biocontainment platform will be useful alone in environments with naturally abundant fluoride, the platform offers great utility in combination of existing and emerging biocontainment mechanisms.

Line 90: I don't think that mentioning the maximum NaF concentration measured is very helpful here and would prefer the authors talk about the media.

We agree with the Reviewer and have removed the text mentioning the maximum measured concentration of fluoride. The figure caption now reads,

“(b) Dose response curve of the biocontainment (i.e. knockout) strain illustrates three orders of magnitude greater sensitivity to fluoride compared to the parent wildtype strain.”

Line 93: Figure 2f should be extended to show OD(600) for 100, 50 and 25 μ M to show the relevant range of ground water fluoride concentrations.

We thank the Reviewer for this suggestion. We wish to note that we are careful to point out the limitations of the biocontainment strain at low fluoride concentrations. As we state on lines 76 - 80,

*“Thus, while fluoride sensitivity may serve as a robust biocontainment measure in areas known to have high fluoride concentrations, **care should be taken to evaluate local fluoride abundance**. Accordingly, fluoride-sensitivity enables passive biocontainment wherein cellular fitness is unperturbed under standard laboratory conditions and markedly reduced **in nature where fluoride is in sufficient abundance**.”*

We can infer from the data presented in Fig. 2b,f that the biocontainment strain is expected to grow with reduced fitness in the presence of 25 – 100 μ M fluoride. As such, while fluoride-sensitivity may result in reduced cellular fitness at these levels of fluoride, we are careful to point out that this biocontainment mechanism is robust at high fluoride concentrations. Since our recommendation is to implement fluoride-based biocontainment at high fluoride concentrations, we believe that the additional data points at fluoride concentrations recommended by the Reviewer will not change the conclusions of the results.

While we are happy to consider additional experiments that may challenge our existing conclusions, at this time we are hesitant to conduct experiments that will not change the message of the study as detailed above.

Figure 2f: In their response to my previous question about the difference in WT OD between the left and the right panel the authors state:

"The WT strain was also treated with fluoride (16 mM) prior to washing and resuspension in fresh YPD. Overnight incubation of the WT strain with this concentration of fluoride appears to affect the strain's growth, albeit to a lesser extent compared to the knockout strain."

That is an important observation and should be stated in the manuscript as well.

We agree that this observation is useful to provide a clear description of the results in Fig. 2f, and we have added this observation to the manuscript to lines 118 – 120.

Line 109: Strain escape rate should also be evaluated for 100, 50 and 25 μ M to show the relevant range of ground water fluoride concentrations.

We thank the Reviewer for their attention to detail and experimental design. As noted in a previous response, we explicitly describe the limitations of the biocontainment strain at low fluoride concentrations on lines 76 – 80.

In combination with the data presented in Fig. 2b,f, our conclusion is to recommend consideration of the biocontainment platform in areas where natural fluoride is abundant as stated on lines 76 – 80, and now on lines 240 – 241,

“Third, in areas where fluoride is naturally abundant in the environment, escapees are subject to lethal conditions without manual intervention.”

And on lines 251 – 252,

“In areas where fluoride is naturally abundant, the biocontainment platform is useful to reduce the fitness of GMOs that would otherwise prosper upon release into the environment...”

We would be happy to consider additional experiments if those experiments would significantly alter the conclusions of this manuscript. However, at this time we are hesitant to conduct experiments that will not change the message of the study, especially given the highly restrictive laboratory access limitations that are in place at UCSB that would prevent us from conducting these experiments for several months.

Line 183: FACS signal for the auxotrophic vectors should be included as a positive control and direct comparison (especially pYC). Is the improved signal for pEFEX3 comparable to pYC? Or is pYC still the better option? Figure 3e should also contain a bar for pYC for direct comparison.

We appreciate the suggestion of this additional control to our experiment. We do note that the pCFEX and pEFEX constructs were constructed directly from the control constructs noted by the Reviewer, pYC A₂aR-GFP and pYES A₂aR-GFP. We have previously published a detailed characterization of these constructs in yeast, and refer to these results during discussion of the pCFEX and pEFEX constructs on lines 173 – 182,

“The FEX vectors display varying degrees of similarity to their auxotrophic counterparts characterized previously³³. Expression of A₂aR-GFP from the high-copy backbones, pEFEX1 and pYES, yields similarly low fluorescence intensities, and the majority of each population displays fluorescence intensities comparable to autofluorescence. In contrast, A₂aR-GFP expression from the auxotrophic low-copy vector, pYC, results in significantly greater MFI over background compared to the analogous FEX vector. Additionally, cells harboring pYC A₂aR-GFP exhibit a bimodal fluorescence distribution, whereas a unimodal distribution is associated with pCFEX1 A₂aR-GFP where the majority of cells exhibit fluorescence intensities indistinguishable from autofluorescence. In all, A₂aR-GFP expression from the FEX vectors was initially poorer than expression from their auxotrophic counterparts.”

We understand the desire to include controls in independent experiments here; however, we also note the difficulty to undertake experiments during this time of extremely limited laboratory access. Especially given that we have referenced previous related work to address this issue, we respectively decline to repeat this control for this work.

Line 205: To my previous question the authors state:

"While we agree that the low copy pCFEX plasmid system was not successfully optimized for A₂aR-GFP production in this study, we stress that membrane proteins are particularly challenging to produce in a heterologous host. We chose this protein to challenge the FEX plasmid systems, and we find that expression of A₂aR-GFP from each vector results in increased MFI compared to a negative control (Fig.3c), which suggests that the selection marker is functional, albeit to an suboptimal extent."

It would be helpful to see the results for a less stringent/taxing expression, maybe just GFP alone. This would help convincing me that the low copy number plasmid works and that the unsuccessful experiment with A₂aR-GFP is due to the complex nature of the POI.

We agree with the Reviewer that the expression of GFP from the pCFEX and pEFEX backbones would be interesting and test the hypothesis we proposed. Although we have sought to clone these constructs previously, only the pEFEX GFP construct was cloned. Unfortunately, the pCFEX GFP construct, the more interesting case, was not obtained after repeated attempts.

Again, we note the difficulty to undertake experiments during this time. Given the current situation, and the fact that we do not believe this experiment would alter the message of the manuscript, we respectfully decline this request.

Line 238: Abundant in the environment at inhibiting concentrations?

We have modified the text in line 238, now line 240, to read,

“Third, in areas where fluoride is naturally abundant in the environment, escapees are subject to lethal conditions without manual intervention.”

Reviewer #2 (Remarks to the Author):

Summary:

This manuscript is a revised version of one that I recently reviewed for a sister journal. The authors have satisfactorily addressed the major and minor critiques that I described in my original review. I recommend the paper for publication in *Nature Communication*.

We thank the Reviewer for their consideration and review of our article, and we are pleased that it has been deemed suitable for publication in *Nature Communications*.